# Extracellular Vesicles in Diffuse Large B Cell Lymphoma: Characterization and Diagnostic Potential

**DOI:** 10.3390/ijms232113327

**Published:** 2022-11-01

**Authors:** Rune Matthiesen, Paula Gameiro, Andreia Henriques, Cristian Bodo, Maria Carolina Strano Moraes, Bruno Costa-Silva, José Cabeçadas, Maria Gomes da Silva, Hans Christian Beck, Ana Sofia Carvalho

**Affiliations:** 1Computational and Experimental Biology Group, NOVA Medical School-Research, Faculdade de Ciências Médicas, Universidade NOVA de Lisboa, Campo dos Mártires da Pátria, 130, 1169-056 Lisbon, Portugal; 2Instituto Português de Oncologia, Departament of Hematology, 1099-213 Lisbon, Portugal; 3Champalimaud Physiology and Cancer Programme, Champalimaud Foundation, 1400-038 Lisbon, Portugal; 4Centre for Clinical Proteomics, Department of Clinical Biochemistry, Odense University Hospital, DK-5000 Odense, Denmark

**Keywords:** diffuse large B cell lymphoma, extracellular vesicles, cancer, diagnostic, liquid biopsy, plasma

## Abstract

Diffuse large B cell lymphoma (DLBCL) is an aggressive B cell lymphoma characterized by a heterogeneous behavior and in need of more accurate biological characterization monitoring and prognostic tools. Extracellular vesicles are secreted by all cell types and are currently established to some extent as representatives of the cell of origin. The present study characterized and evaluated the diagnostic and prognostic potential of plasma extracellular vesicles (EVs) proteome in DLBCL by using state-of-the-art mass spectrometry. The EV proteome is strongly affected by DLBCL status, with multiple proteins uniquely identified in the plasma of DLBCL. A proof-of-concept classifier resulted in highly accurate classification with a sensitivity and specificity of 1 when tested on the holdout test data set. On the other hand, no proteins were identified to correlate with non-germinal center B-cell like (non-GCB) or GCB subtypes to a significant degree after correction for multiple testing. However, functional analysis suggested that antigen binding is regulated when comparing non-GCB and GCB. Survival analysis based on protein quantitative values and clinical parameters identified multiple EV proteins as significantly correlated to survival. In conclusion, the plasma extracellular vesicle proteome identifies DLBCL cancer patients from healthy donors and contains potential EV protein markers for prediction of survival.

## 1. Introduction

B-cell lymphomas develop from abnormal B-lymphocytes and account for 85% of all non-Hodgkin’s lymphomas (NHLs). Diffuse large B-cell lymphoma (DLBCL) is the most common form of lymphoma, accounting for up to 40% of all newly diagnosed cases of NHL [1]. As a heterogeneous entity, DLBCL is also characterized by a variety of morphologic variants, diverse molecular abnormalities, and variable clinical behaviors and therapeutic responses [2,3]. It can arise de novo or as a transformation from a low grade B-cell malignancy, such as follicular lymphoma or chronic lymphocytic leukemia. The disease demonstrates remarkable progression-free survival and overall survival in clinically and biologically defined subgroups of patients. Unfortunately, approximately 40% of patients will relapse or develop refractory disease. Simple clinical characteristics, such as age, lactate dehydrogenase levels, the number and sites of metastases involvement, the stage, and performance status can be scored in a manner that stratifies R-CHOP (rituximab, cyclophosphamide, doxorubicin, vincristine and prednisone) treated DLBCL patients in groups with varying survival [4]. Gene expression profiling (GEP) studies of DLBCL revealed two main sub-types: germinal center B-cell like (GCB) and activated B-cell like (ABC). In addition, a rarer primary mediastinal B-cell type was identified. Recently, deeper molecular analysis of large cohorts of DLBCL newly diagnosed biopsies revealed different molecular subtypes that were associated with responses to immunochemotherapy [5,6]. Non-GCB tumors, defined by immunohistochemistry algorithms, include the unclassifiable and the ABC DLBCL subtype; the latter is associated with poor treatment outcomes. The gold-standard method for DLBCL gene expression sub-typing is the Affymetrix method based on frozen tissue (FT) [7,8,9]. This severely limits the ability to utilize this assay in routine clinical practice, as fresh tissue is not routinely collected during patient diagnosis. The Lymph2Cx assay, a digital 20-gene expression based test (NanoString) for cell of origin (COO) assignment requiring formalin-fixed paraffin-embedded tissue (FFPET), has demonstrated very high rates of accurate assignment when tested against the gold-standard FT method [10]. Despite good segregation between ABC and GCB subtypes, both methods rely on RNA, which is susceptible to degradation by chemical treatments such as those used in FFPET preparation and requires fresh tissue as a source. COO assignment in standard practice applying immunohistochemical methods (BCL6/CD10/MUM1) in commonly available FFPET is less precise but relatively inexpensive. Attempts to develop multiple immunohistochemistry (IHC)-based algorithms such as Hans, Tally, Choi and Visco-Young suffered from lower than desired agreement rates with GEP classifications, and required staining of four to eight sections of limited biopsy material [1,11].

Success of targeted therapies for DLBCL relies on minimally invasive, robust strategies that segregate patients into molecular groups with high reliability. Liquid biopsy is a minimally invasive method that potentially provides information on circulating materials such as cell-free DNA (cfDNA), circulating tumor cells (CTCs), and extracellular vesicles (EVs) to detect molecular alterations that are indicative of cancer progression.

Using PCR technology, circulating tumor cells in the blood have been reported to be associated with recurrence in patients with indolent lymphoma and mantle cell lymphoma. Unlike indolent lymphomas, DLBCLs are associated with a relatively low level of circulating tumour cells. Next-generation sequencing (NGS) applied to cell-free tumour DNA in the serum of patients with DLBCL detected clonal immunoglobulin gene sequences, which correlated with poor survival [12].

Since the discovery of EVs, including exosomes, in human body fluids, compelling research studies have been published exploiting the potential of EVs as a source of stable biomarkers. EVs are cell-derived vesicles composed of lipid bilayers, constituting a heterogeneous vesicle population of different sizes, biogenesis and origins. These structures shed from all cell types, retaining part of the donor cell signature as well as specific EV-related factors [13]. EVs have been described as being involved in cell-cell communication in various physiological processes, such as antitumoral immune responses, and conferring resistance to therapy [14]. Additionally, EVs functionally operate at a systemic level and contain circulating targets for diagnosis, prognosis, and potential treatment targets. Given that EVs have been recognized as a promising source of diagnostic biomarkers, various attempts to establish methods to detect disease-specific circulating EVs have been reported. Recently, the proteome of tumor-derived EVs in cancer and non-cancer patients’ plasma has been shown to discriminate cancer from non-cancer and to identify proteins with potential to serve as prognostic biomarkers for melanoma [15]. A pan cancer signature of plasma EVs that extended to several types of cancer, allowing early-stage detection, treatment response, and potentially for diagnosing tumors of unknown primary origin, was proposed [16]. EVs present the advantage that the biomolecules are encapsulated in a lipid bilayer, preventing attack by free nucleases and proteases. In addition, for mass spectrometry-based proteomics, EVs serve as an efficient method to remove abundant proteins. EVs are also readily available from a standard blood sample enabling longitudinal monitoring. In this study, we explored the proteome of plasma EVs from newly diagnosed DLBCL patients and age-matched healthy donors (HD) by state-of-the-art high-throughput mass spectrometry. The results obtained can be explored in the future for accurately identify minimal residual disease (MRD) as well as predicting patient survival using EV proteomics. MRD monitoring could help identify early relapses or treatment failure.

## 2. Results

The present study aimed at comparing the proteome of vesicles isolated from the plasma of DLBCL patients and healthy donors as well as between DLBCL patients. Patients’ characteristics are summarized in Table 1.

### 2.1. Patient Characteristics

This study included 32 patients diagnosed with DLBCL (Table 1) and fifteen age-matched healthy donors. Twenty-one patients were classified as GCB, ten as non-GCB and one case was unclassified by Hans immunohistochemistry algorithm [17]. Twenty-six patients presented a complete response, while the remaining presented partial responses, progressive disease, or had died due to disease progression.

### 2.2. Plasma EVs of DLBCL Patients and HDs

The present study’s primary aim is to compare the proteome of extracellular vesicles isolated from the plasma of DLBCL patients and HDs. For mass spectrometry analysis, we have isolated extracellular vesicles from 32 newly diagnosed DLBCL patients and 15 aged matched HDs using sequential centrifugation, as previously described [18]. The particles have been analyzed by nanoparticle tracking analysis (Figure 1a,b). EVs plasma were characterized by different size subpopulations enriched in the 100 nm range. However, the comparison of particles size distribution between cancer and HDs showed a more heterogeneous population for lymphoma patients compared to healthy donors (Figure 1c,d).

The particle size of plasma EVs samples from patients showed a significant increase compared to those from HDs. Notably, the size range of particles in the EV preparations from DLBCL patients varied between 90–140 nm while with the exception of a single individual, particles from HDs varied between 100–110 nm (Figure 1). Subgrouping according to prognosis (survival and deceased patients) showed no significant differences (data not shown).

### 2.3. EV Protein Markers

We previously proposed an EV quality control software tool based on mass spectrometry data [14,19]. The tool compares the expression of EV markers frequently reported in the literature and the top ten most abundant exosome markers from a publicly available EV database (ExoCarta) [20]. This tool was further developed in the present study to also include markers for large micro vesicles (RGAP1, MIC60) and furthermore highlight widely accepted exosome markers such as syntenin (SDCBP), ALIX (PDCD6IP) and CD63 (Figure 2). Two alternative isolation methods of EV enrichment were used in the reference data for EVs from cell lines: differential ultra-centrifugation [16] and PEG-based precipitation [21]. All markers are present in the plasma EVs, although at a slightly lower intensity compared to EVs isolated by similar methods from cell line conditioned media and bronchoalveolar lavage (BAL). EV isolation from plasma is notoriously difficult and known to contain platelet EVs [22]. Markers indicating contamination from other subcellular fractions such as the endoplasmic reticulum (CANX), Golgi apparatus (GOLGA2), mitochondria (BCL2), and nucleus (NUP98) were absent or in lower range compared to previous studies. Microvesicles markers were not observed by any of the methods and sample sources included (Figure 2).

### 2.4. Qualitative Comparison of EV Proteome from DLBCL Patients and HDs

Plasma EV proteomes from DLBCL patients and HDs were analyzed by mass spectrometry. Across a total of 98 liquid chromatography tandem mass spectrometry (LC-MS/MS) runs, 2595 protein isoforms were identified. The protein isoforms were mapped to their corresponding genes, resulting in a final number of 1197 proteins. Figure 3a depicts the number of proteins identified factored on cancer status. The DLBCL group displayed a high number of identified proteins, although there were considerably fewer samples from the HD group compared to the DLBCL group. The DLBCL group also displayed more unique proteins compared to the HD group (Figure 3b). The unique proteins for the DLBCL and HD groups are listed in Appendix A. According to KEGG pathway analysis (Figure 3c), proteins identified in DLBCL were more enriched in proteasome, infection related functions, antigen presentation, and glycolysis and gluconeogenesis functions than those from the HD group. Complement and coagulation cascades and systemic lupus erythematosus were found significantly enriched in proteins identified from both the DLBCL and the HD group. We used Gene Ontology (GO) (http://www.geneontology.org/, accessed on 15 August 2022) to perform enrichment analysis based on the Gene Ontology–Biological Process (GO-BP), Cellular Component (GO-CC) and Molecular Function (MF) annotations. Figure A1, Figure A2 and Figure A3 represent the gene ontology functional enrichment analysis of all identified proteins from the DLBCL and the HD group. This enrichment analysis indicated that the EV plasma proteome of patients and HDs were particularly enriched in GO-CC associated with extracellular exosomes with 548 (DLBCL) and 362 (HD) proteins classified in this category, confirming the quality of the EVs isolation method used (Figure A1). EV plasma proteome included the described protein cargo with tetraspanins membrane organizers, CD9, CD81, CD82, CD63, intracellular trafficking components such as RAB, GTPases, annexins and chaperone molecules from the HSP90 and HSP70 protein families (Appendix A). As shown in Figure A2, the molecular functions associated with the proteome of all identified proteins in plasma EVs were threonine-type endopeptidase activity, structural constituent of cytoskeleton, GTPase activity (mainly DLBCL) and antigen binding (mainly HD). The most significant enrichment in GO-BP associated with innate immune response and platelet activation, with 79 (HD)/105 (DLBCL) and 45 (HD)/67 (DLBCL) proteins identified in each category, respectively (Figure A3). Additionally for GO-BP, negative regulation of endopeptidase activity was mainly identified in HDs (Figure A3). GTPase activity, antigen processing related functions, and ubiquitin ligase activity in cell cycle were mainly enriched in samples from DLBCL patients (Figure A3).

To evaluate if the protein richness is indeed higher in the DLBCL group, a subsampling of 100 permutations was drawn, assuming a different number of samples. The total number of non-redundant proteins for each sample number was counted, and standard error was calculated for each draw (Figure 4). This modeling provides a view of protein richness given the measurement strategy in the two sample groups. The estimated protein richness at the plateau by Jack1 method was 973 (45), and it was 1291 (53) for the HD and DLBCL groups, respectively (Figure 4b, gene collapsed proteins).

The statistical difference in number of protein identifications was estimated based on 50 random subsamples of 10 HD and 10 DLBCL sample groups (Figure A4) and the result indicates a highly significant difference in the number of proteins identified in the two sample groups. Overall, our analysis suggests that DLBCL samples contain a more diverse proteome and the variance among the number of identified proteins in the DLBCL group is higher than in the HD group (e.g., variance depicted in Figure 4 and Figure A4).

### 2.5. Quantitative Analysis of Differential Regulated EV Proteome

The label-free quantitative proteomics data were obtained in two batches. To correct for batch effects, MS data was analyzed separately for principal component analysis (Figure A5) and considered in subsequent statistical analysis using the limma R package. In total, 1193 (DLBCL versus HD) and 1183 (non-GCB versus GCB) proteins were identified and quantified after collapsing into the encoded genes and removing low expression proteins. A statistical comparison of quantitative proteomics data between DLBCL and HD group (Appendix A) and between non-GCB and GCB (Appendix A) revealed significant differences in 264/402 expressed proteins from DLBCL versus HD (adj. *p*-value < 0.05/*p*-value < 0.05). For non-GCB and GCB 51 differentially expressed proteins (DEPs) were identified (*p*-value < 0.05).

Volcano plots showing the comparison between DLBCL and HD, and between non-GCB and GCB proteomes, are depicted in Figure 5a,b, respectively. Non-GCB versus GCB resulted in less DEP in relation to DLBCL versus HD comparison (Figure 5a,b). Given that platelets are considered a major contaminant in EV preparations, potential contaminant proteins from erythrocytes, platelets and the blood coagulation system extracted from Geyer et al. [23] are highlighted in red, orange and blue in the figure, respectively. Despite the fact that platelet annotated proteins are present in the plasma EV samples, minimal overlap of potential contaminant proteins with significant DEPs was observed (Figure 5a,b). Immunoglobulins and guanine nucleotide-binding protein family members were up-regulated between DLBCL and HD (Figure 5a). The main immunoglobulin proteins are characterized by the specific rearrangements in the immunoglobulin kappa and lambda light chains such as IGKV3-20, IGKV3D-11 and IGLV1-40. The G proteins GNA12/GNA13 as well as GNAI1/3 were the main signal transducers overexpressed in DLBCL, while SERPINF1, VNN1, CLHC1 and HLA-A were downregulated in the EV plasma of DLBCL. In the GCB group, plasma EVs are enriched in histone proteins such as histone H2B as well as in heat shock protein TRAP1 and the glycosyltransferase ST3GAL6 (Figure 5b).

The results of the functional enrichment analysis of differentially expressed proteins are depicted in Figure 6 and summarized in Appendix A. Functional analysis of DEP for the comparison of DLBCL versus HD showed proteins involved in lupus related systemic inflammation (Figure 6a), platelet activation (Figure 6b), regulation of glycoprotein metabolic processes (Figure 6b), and proteins involved in GTPase activity (Figure 6b,c). Deregulated platelet activation (Figure 6b and Appendix A) proteins were mainly up-regulated in DLBCL. Note that platelet contaminant markers described by Geyer et al. [22] are distinct from the functional category of platelet activation. In addition, within the GTPase activity category, there were significantly up-regulated and down regulated proteins in DLBCL (Figure 6c). The cellular location of deregulated proteins included the extracellular space, blood microparticles, and the nucleosome and complement complex (Figure 6d).

### 2.6. DLBCL vs. HD Classification

A partial least square (PLS) regression model was built, given the large number of significantly regulated proteins when comparing DLBCL and HD cases. A 66% split, ensuring a balanced training set, was applied to divide the data into training and test sets (Figure 7a). The PLS model tested on the test data displayed excellent performance based on receiving operating characteristics (ROC), with an area under the curve (AUC) of 1 (Figure 7b). The confusion matrix revealed zero misclassifications out of 15 cases (Figure 7d). The model was also built without enforcing a balanced training set, which again resulted in perfect classification without any misclassifications. Caution must, however, be highlighted due to the low number of test samples available. For a larger test set, some errors would be expected. However, given the large proteome differences between EVs from DLBCL patients and HD, we expect that this classification based on the plasma EV proteome will present high accuracy in practice. The most important protein features for the model are represented in Figure 7c. Immunoglobulin proteins appeared with the highest importance score for the model closely followed by G protein GNAI3. This model is highlighted as a proof of concept to demonstrate the feasibility of classifications based on plasma EVs. However, it must be optimized on a larger dataset, which may change the optimal proteins and classification algorithm.

### 2.7. Survival Analysis

Univariate Cox proportional hazards regression models were fitted to the observed survival, proteomics and clinical data for all 32 DLBCL patients in the cohort. The proportional hazard (PH) assumption was assessed using Schoenfeld residuals [24]. The Cox–Mantel log-rank tests for the top proteins (*p*-value < 0.005 and Cox–Mantel log-rank tests < 0.01) are listed in Table 2. None of the measured clinical variables correlated significantly with survival when the above thresholds were applied. In addition, the total number of proteins identified in each case was also not correlated with survival. Figure 8 depicts the Kaplan–Meier plots for immunoglobulin lambda constant 1 (IGLC1), immunoglobulin lambda like polypeptide 5 (IGLL5), proteasome subunit beta type-2 (PSMB2) and coronin-1A (CORO1a), which are among the most significant predictors for survival.

## 3. Discussion

Blood circulating EVs can reflect changes occurring in different tissues and organs. The abundance and population diversity of EVs in plasma is still a matter of debate. The quantified particle concentration in the EVs plasma fraction from DLBCL patients and HDs showed that cancer patient-derived plasma contained a significantly higher number of particles compared to that from HDs and a broader range of particle size mode (Figure 1). This has also been observed by others reporting a clinical significance for other types of cancers, such as, for example, glioblastoma [25]. To access EV plasma composition, we have presented here a quality assessment tool using the most reported proteins publicly available in EV database (ExoCarta) [20,26,27] based on MS-based proteomics studies of EVs isolated from the conditioned cell medium (CCM) by different isolation methods. The marker expression in the current study is then compared with marker expression from previous large EV studies. EVs from cell lines, body fluids such as bronchoalveolar lavage fluid, and plasma studies were included. DLBCL-plasma EVs were enriched in tetraspanins CD9, CD63 and CD81, and enriched in classical EV markers generated via the endocytic pathway (Figure 2). Proteins involved in EV biogenesis [13] such as PDCD6IP, FLOT1 and TSG101 were also identified, although at a lower expression level compared with the tetraspanins proteins. When compared to CCM-EVs, the protein level of EV markers was higher in CCM-EVs compared with plasma EVs (Figure 2). Isolation of plasma EVs is normally restricted to a few milliliters of sample compared with an almost infinite amount of CCM potentially available. On the other hand, the diversity of EV subpopulations and other particles in plasma surpasses that of CCM, which normally originates from a single cell line. Nevertheless, when the ranked expression of tetraspanins and PDCD6IP, FLOT1 and TSG101 is compared between CCM and plasma derived EV samples, our study suggests that the proteins identified are enriched in extracellular vesicles of the endocytic pathway. However, the presence of vesicles generated via the secretory pathway in plasma cannot be discarded. Indeed, GO functional enrichment analysis of cellular components of identified proteins among DLBCL and HD groups are equally enriched in cellular component categories such as extracellular region, extracellular vesicular exosome, extracellular space, proteasome complex and blood microparticle (Figure A1). Nevertheless, in cancer patients, proteins in cellular component categories related to vesicle and focal adhesion are more enriched compared to those from HD (Figure A1). In our study, CD9 and CD81 were 2.2- and 5.2 log_2-_ fold significantly up regulated (adj. *p*-value < 0.001) between the DLBCL and HD group, which suggests that the increased particle concentration in plasma comes from vesicles with CD9 and CD81 markers. A significant increase in CD9-CD63 EVs was also observed in the plasma of colorectal cancer patients compared to age-matched HDs [28]. Blood can, in principle, contain a variety of EVs released from many different cell types, but studies demonstrate that the main source of EVs in HD plasma originates from platelets (primarily) and erythrocytes [29]. On the other hand, tumour cells secrete an increased number of EVs, which have been described to contain molecules capable of transferring oncogenic traits to the neighboring cells or to disseminate to distant organs. In our study we observed that the proteome of DLBCL plasma EV contain unique proteins which can derive from the tumor itself or from cells involved in the systemic response (Appendix A). The accumulation of exosomes carrying tissue factor (TF) in plasma has been linked to venous thrombolytic events, which are common in hematologic malignancies [30].

TF can interact with components of the coagulation cascade, and the large numbers of pro-coagulant vesicles in the plasma of cancer patients is considered a prognostic factor for dismal outcome [30]. Furthermore, it should be noted that, as proposed by other authors, immune cells probably secrete most EVs in lymph nodes or locally at the required site and, thus, only low levels of these particles would be present in the circulation [31]. These examples illustrate the variety of studies that are being carried out in the cancer field and the promising potential of EVs. Nevertheless, they also highlight the long path ahead.

With regard to plasma EVs, our study demonstrated that it reflected a pathological condition since 1151 proteins have been identified in EV plasma from DLBCL patients compared to 832 in HD EV plasma samples from HD. Even though the sampling between DLBCL and HD groups was unbalanced, estimation of protein richness verified a highly significant difference in the number of proteins identified in the two sample groups (Figure 3). This significant higher proteome complexity in malignant samples is in line with our previous findings from bronchoalveolar lavage samples [19,32]. Regarding protein quantification, 264/402 proteins were significantly regulated (adj *p*-value < 0.05/*p*-value < 0.05) between DLBCL patients and HDs. Among the significantly regulated proteins were some related to innate immune response. More specifically, these included immunoglobulins and proteins binding to G protein–coupled receptors such as guanine nucleotide-binding proteins, GNA12/13 (Figure 5a and Figure 6). B-cell lymphomas are characterized by somatic hypermutation (SHM) events in immunoglobulin genes that lead to abnormal SHM; IGV SHM may enhance the B-cell receptor (BCR) affinity and B-cell survival, suggesting unfavorable prognostic effects [33]. Young et al. demonstrated that DLBCL tumor cells bear rearranged IgVH segments that are specific for ABC subtype tumors, and its viability depends on autoantigens [34]. Recently, we have performed a comprehensive proteomics profiling of EVs isolated from a panel of DLBCL cell lines, including GCB and ABC-like subtypes. In fact, the B-cell receptor pathway was significantly enriched in EVs secreted from DLBCL-ABC subtype cells [14]. G proteins are signaling transducers, and although they don’t mediate directly signaling cascades triggered by the antigen activation of immune cells, G proteins might have important modulatory roles. For example, Gi protein signaling inhibition enhanced the capacity of splenocytes to produce IL-12 in response to both microbial and nonmicrobial stimuli.

G-proteins are up-regulated in GCB cells, with *Gna13* transcripts appearing more abundantly [35]. EVs have been implicated in several functions, such as in adaptive immune response to eliminate tumor cells [31]. The H2B protein family was regulated between non-GCB and GCB groups, and its detection in plasma has been associated with neutrophil extracellular traps (NETs) as well as vesicles [13]. Interestingly, early reports on NETs describe the antimicrobial activity of histones [36].

Among the proteins differentially regulated between DLBCL and HD groups, when DLBCL subtypes were considered, there was a clear enrichment for proteins in the Systemic Lupus erythematosus (SLE) KEGG pathway (Figure 6). SLE is an autoimmune disease in which immune complexes comprised of autoantibodies and self-antigens are deposited, particularly in the renal glomeruli, and mediate a systemic inflammatory response by activating complement or via Fcgamma receptor (FcgR)-mediated neutrophil and macrophage activation. In fact, the risk of SLE in hematological cancers, especially in non-Hodgkin’s lymphoma, is increased approximately four-fold [37], and this has been postulated to be due to both immunological abnormalities and drug exposures as well as viral origin. Pentraxin 3 (PTX3) was downregulated in the EV fraction plasma of DLBCL patients. PTX3 can be induced by different soluble factors such as proteins from the interferon family, interleukins and tumour necrosis factors [38]. The role of PTX3 in cancer is not clear, and its contribution as a tumour promoter or suppressor is being equally reported. In multiple myeloma (MM), PTX3 affects in vitro and in vivo fibroblast growth factor 2-mediated MM angiogenesis, promoting cytotoxicity in MM cells by inhibiting the cross-talk between plasma cells, endothelial cells and fibroblasts [39]. On the other hand, it has been demonstrated in PTX3 deficient mice that tumorigenesis is promoted and is mediated by complement-dependent tumor-promoting inflammation [40].

Single cell analysis using CyTOF demonstrated that DLBCL tumor cells are phenotypically unique to each patient, with a small number of samples displaying distinct sub-clonal structures, often distinguished by the differential expression of immune-related proteins such as MHC-II rather than GCB vs. ABC [41]. However, this might be a result of the limited number of samples. However, several studies [42] have described poor agreement with non-GCB and GCB classification for prognosis, suggesting that in DLBCL, systemic components have the potential to contribute to refining the subclassification of this most prevalent type of NHL. In fact, we identified multiple protein markers associated with survival, such as IGLC1, IGLL5, PSMB2 and CORO1a (Table 2). All of these potential protein markers have previously been associated with poor prognosis in other cancers [43,44,45,46]. It is possible that IGLC1 and IGLL5 stems from co-isolated immune complexes rather than EVs or may also be associated with the membrane of the EVs. PSMB2 is part of the proteasome complex, which is typically upregulated in many cancers. Unfortunately, inhibition of the proteasome with molecules such as bortezomib [47] and lenalidomide [48] in relapsed/refractory DLBCL has demonstrated modest responses. However, PSMB2 is not targeted by bortezomib and lenalidomide and may constitute a potential target. CORO1a plays a role in recurrent viral infection and is also associated with a poor survival rate, having a positive correlation with lymph node metastasis in gastric cancer patients [49]. Additionally, the present results demonstrate excellent separation for DLBCL patients compared to HDs, suggesting that EV proteins or a subset thereof may be useful for diagnosis or monitoring risk of recurrence. The model presented here is considered a proof-of-concept and should be tested on a larger cohort and compared with other clinical parameters already proven to have diagnostic value. Finally, this study opens a new research line aimed at understanding diversity in the EV proteome of DLBCL patients and its impact on therapeutic response.

## 4. Materials and Methods

### 4.1. Plasma of Cancer Patients and Healthy Donors

We have collected blood from DLBCL patients at diagnosis, prior to treatment, and of age-matched HDs, and separated the blood into blood cells, platelets and plasma. Venous blood samples were obtained from patients with DLBCL that were receiving treatment at the Portuguese Institute of Oncology of Lisbon Francisco Gentil, as well as from healthy volunteers. Blood samples were collected at diagnosis at a single center between December 2017 and January 2020. All patients were treated with R-CHOP therapy or related regimens (R DAEPOCH, RCEOP), with the exception of one patient who received palliative care. On average, the treatment period lasted between three to five months, with a median follow-up of 1 year. All patients received R-CHOP or R-CHOP-like chemotherapy with curative intent. At the time of diagnosis, only one patient presented with central nervous system involvement. Peripheral blood was collected from newly diagnosed DLBCL patients prior to treatment and from age matched healthy volunteers into EDTA lavendertop tubes. Collection tubes were centrifuged at 200× *g* for 10 min at RT. The top layer was centrifuged at 1000× *g* for 10 min to collect the plasma. Plasma was aliquoted into 2 mL vials and stored at −80 °C until further use.

### 4.2. Isolation of Extracellular Vesicles from Human Plasma

As previously described [18], frozen plasma specimens were thawed and centrifuged at 3000× *g* for 20 min at 4 °C and then at 12,000× g for 60 min at 4 °C. Clarified plasma was ultracentrifuged in an Optima TM L-80XP ultracentrifuge (Beckman Coulter, Brea, CA, USA) at 100,000× *g* at 4 °C for 120 min with a Type 50.2 Ti rotor (k-factor: 157.7) to pellet EVs. The supernatant was carefully removed, and crude EV-containing pellets were resuspended in ice-cold PBS, followed by floatation on a sucrose cushion (30%, D_2_O) (Sigma-Aldrich, Saint Louis, MO, USA) for 60 min at 100,000× *g* at 4 °C to remove non-EV protein complexes. After washing, by pelleting the EVs collected in the sucrose cushion for 16 h at 100,000× *g* at 10 °C, EVs were resuspended in phosphate-buffered saline (Sigma-Aldrich, Saint Louis, MO, USA) and subjected to nanoparticle tracking analysis using NanoSight NS300 equipment (Malvern Instruments, Inc., Westborough, MA, USA).

### 4.3. Protein Measurements

Protein concentrations in isolated exosome fractions were measured using a bicinchoninic acid (BCA) protein assay kit (Pierce Biotechnology, Rockford, IL, USA) according to the manufacturer’s instructions.

### 4.4. Peptide Sample Preparation

Samples containing a minimum of 20 μg of total proteins were further processed by the filter-aided sample preparation (FASP) method. In short, protein solutions containing SDS and DTT were loaded onto filtering columns (Millipore, Billerica, MA, USA) and washed exhaustively with 8M urea (GE, Healthcare, Marlborough, MA, USA) in HEPES buffer (Sigma-Aldrich, Saint Louis, MO, USA) as previously described [14]. Proteins were reduced with DTT (BioRad, Hercules, California, USA) and alkylated with IAA (GE, Healthcare, Marlborough, MA, USA). Before trypsin digestion overnight at 37 °C, proteins were equilibrated with ammonium bicarbonate buffer (Sigma-Aldrich, Saint Louis, MO, USA). Protein digestion was performed by overnight digestion with sequencing grade trypsin (Promega, Madison, WI, USA).

### 4.5. Mass Spectrometry Analysis

As previously described [14], samples were analyzed by mass spectrometry-based proteomics using nano-LC-MSMS equipment (Dionex RSLCnano 3000) coupled to an Exploris 480 Orbitrap mass spectrometer (Thermo Scientific, Hemel Hempstead, UK). In brief, samples were loaded onto a custom-made fused capillary pre-column (2 cm length, 360 µm OD, 75 µm ID, flowrate 5 µL per minute for 6 min) packed with ReproSil Pur C18 5.0 µm resin (Dr. Maisch, Ammerbuch-Entringen, Germany), and separated using a capillary column (25 cm length, 360 µm outer diameter, 75 µm inner diameter) packed with ReproSil Pur C18 1.9-µm resin (Dr. Maisch, Ammerbuch-Entringen, Germany) at a flow of 250 nL per minute. A 56 min linear gradient from 89% A (0.1% formic acid) to 32% B (0.1% formic acid in 80% acetonitrile) was applied. Mass spectra were acquired in positive ion mode in a data-dependent manner by switching between one Orbitrap survey MS scan (mass range *m/z* 350 to *m/z* 1200) followed by the sequential isolation and higher-energy collision dissociation (HCD) fragmentation and Orbitrap detection of fragment ions of the most intense ions with a cycle time of 2 s between each MS scan. MS and MSMS settings: maximum injection times were set to “Auto”, normalized collision energy was 30%, ion selection threshold for MSMS analysis was 10,000 counts, and dynamic exclusion of sequenced ions was set to 30 s.

### 4.6. Database Search

The obtained data from the 98 LC-MS runs of 15 HD and 32 DLBCL EV samples were searched using VEMS [50] and MaxQuant [51] (all samples were run with two replicas except one DLBCL and one HD sample that were run as triplicates). The MSMS spectra were searched against a standard human proteome database from UniProt (3AUP000005640). Permuted protein sequences, where arginine and lysine were not permuted, were included in the database for VEMS. Trypsin cleavage allowing a maximum of four missed cleavages was used. Carbamidomethyl cysteine was included as fixed modification. Methionine oxidation, lysine and N-terminal protein acetylation, serine, threonine and tyrosine phosphorylation, ubiquitin diglycine and “LRGG” remnant on lysine were included as variable modifications. A 5 ppm mass accuracy was specified for precursor ions and 0.01 *m/z* for fragment ions. The false discovery rate (FDR) for protein identification was set at 1% for peptide and protein identifications. No restriction was applied for minimal peptide length for VEMS search. Identified proteins were divided into evidence groups as defined by Matthiesen et al. [52].

### 4.7. Statistical Analysis and Machine Learning

Statistical analysis of identified proteins was performed in the R statistical programming language. Protein richness in sample groups given the experimental protocol was estimated using the vegan R package [53]. vegan was also used to plot protein accumulation curves.

Quantitative data from MaxQuant and VEMS were analyzed in the R statistical programming language version 4.04 (The R Foundation, Vienna, Austria). Protein label free quantitation (LFQ) and protein spectral counts from the two programs were preprocessed by removing common MS contaminants followed by a log_2_(x + 1) transformation and removing common MS contaminants. Proteins with an average expression below 5 on the log_2_ scale were filtered out. Protein LFQ values were subjected to statistical analysis utilizing R package limma [54], where the contrast between DLBCL versus HD and non-GCB versus GCB were specified (Appendix A).

Correction for multiple testing was applied using the method of Benjamini & Hochberg [55]. A volcano plot was plotted with ggplot software (The R Foundation, Vienna, Austria) followed by annotating data points with potential contaminant proteins from erythrocyte, platelet and coagulation extracted from Geyer et al. [23]. The R package pROC was applied for plotting receiver operator characteristic (ROC) and calculating the Youden’s J statistic [56].

The partial least squares (PLS) regression models were constructed using the R package caret [57]. The data were log_2_ transformed, centered, and scaled. The training and test data (N = 47) included only data with a definitive diagnosis and no missing data. The data were split to ensure a balanced training set, with 66% of the data used for model training and the remaining 33% used to test the reported performance. Data splitting was constructed to ensure a balanced training set. Highly correlated features with correlation above 0.9 and zero variance features were removed. Before training the PLS classifier, a univariate feature selection based on *p*-values calculated by the limma R package in the training data set were applied. The presented model was restricted to the top 20 features. The training model was optimized using 10-fold repeated cross validation and accuracy as the performance metric.

### 4.8. Functional Enrichment Analysis

Functional enrichment based on the hypergeometric probability test was performed as described previously in R [58,59]. Functional enrichment was based on extracting all functional categories for which at least one of the samples showed a significant enrichment based on the hypergeometric probability test. For these functional categories, the matching proteins’ gene names and numbers of proteins matching the functional categories were extracted and the estimated *p* values were –log_10_ transformed and plotted as heatmaps. Functional enrichment was performed for all identified proteins in each sample group and deregulated proteins when comparing sample groups. Cellular component (CC), biological process (BP), molecular function (MF) and KEGG functional annotation were considered in the analysis. In the case of KEGG enrichment for identified proteins, a filter was applied to only maintain the functional categories with at least 50% unique proteins identified compared to all functional categories with higher significance. The network of how all the top functional categories significantly enriched for KEGG overlap in terms of identified proteins is depicted in Figure A6.

### 4.9. Survival Analysis

Univariate Cox proportional hazards regression models were fitted using the R package survival [60] and the parameter ties were set to “breslow”. Kaplan-Meier plots were constructed with the R package RTCGA [61].

## 5. Conclusions

The results described in this study constitute a proof of concept for the application of proteins EVs as a set of biomarkers for the detection of DLBCL in patient plasma. We demonstrated that the plasma EV fraction was enriched in EV and blood microparticles and could effectively identify DLBCL patients. Classification of DLBCL subtypes based on EVs proteome appeared to be correlated with immune system components such as immunoglobulins proteins or G-protein, but further research on larger patient cohorts is required to confirm this assertion. Moreover, we acknowledge that deeper characterization of plasma EV fraction would improve the sensitivity and specificity of biomarkers. In summary, our data indicate that the protein cargo of plasma EVs from DLBCL patients that reflects the pathological condition might serve as a liquid tumour biopsy upon further validation, and it has the potential to become a proxy of clinical prognosis. Single parameter, multi-parameter and omics approaches can facilitate the identification of reliable cancer biomarkers in EVs, but further work is still required to validate the proposed candidates.

## Figures and Tables

**Figure 1 ijms-23-13327-f001:**
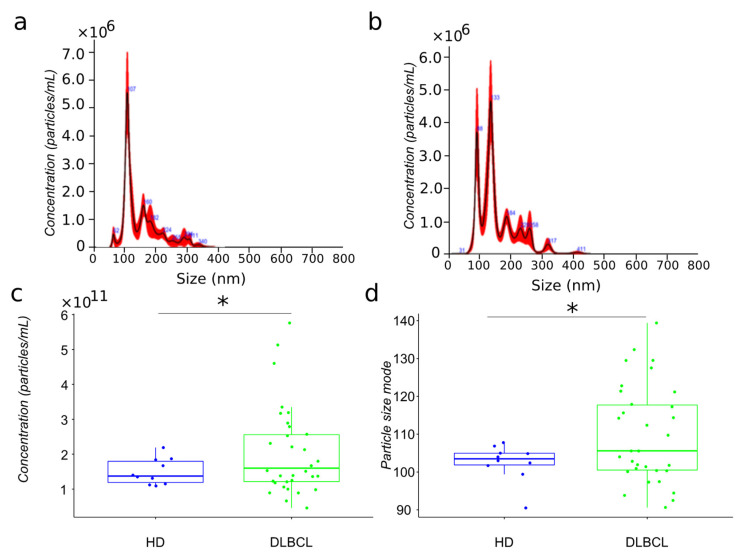
Biophysical characterization by nanoparticle tracking analysis of plasma EV samples from HD and DLBCL groups. Particle size distribution in representative samples of HD and DLBCL groups ((**a**,**b**), respectively). Distribution of estimated particle concentration and particle size mode between DLBCL patients and HDs ((**c**,**d**), respectively). * means significance.

**Figure 2 ijms-23-13327-f002:**
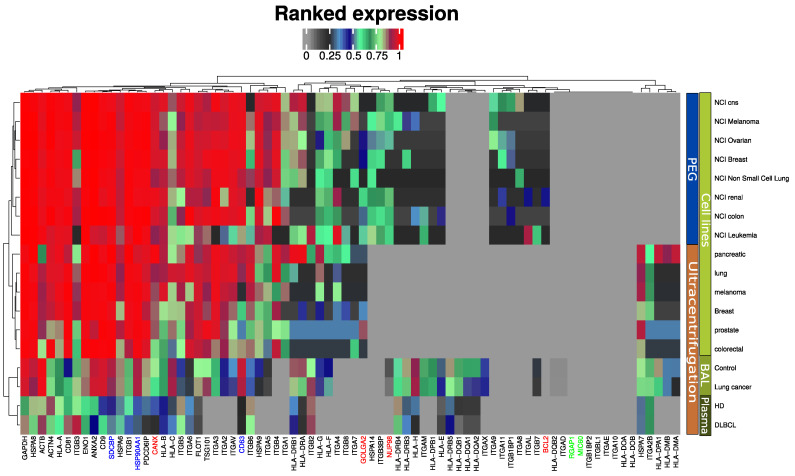
MS-based quantitative comparison between enriched plasma EVs, BAL EVs and cell line isolated EVs of frequently reported exosome protein markers and the ten most abundant exosome markers from a publicly available EV database (ExoCarta). Blue indicates generally accepted exosome markers. Green indicates micro vesicle markers. Red-labeled proteins indicate non-EV proteins.

**Figure 3 ijms-23-13327-f003:**
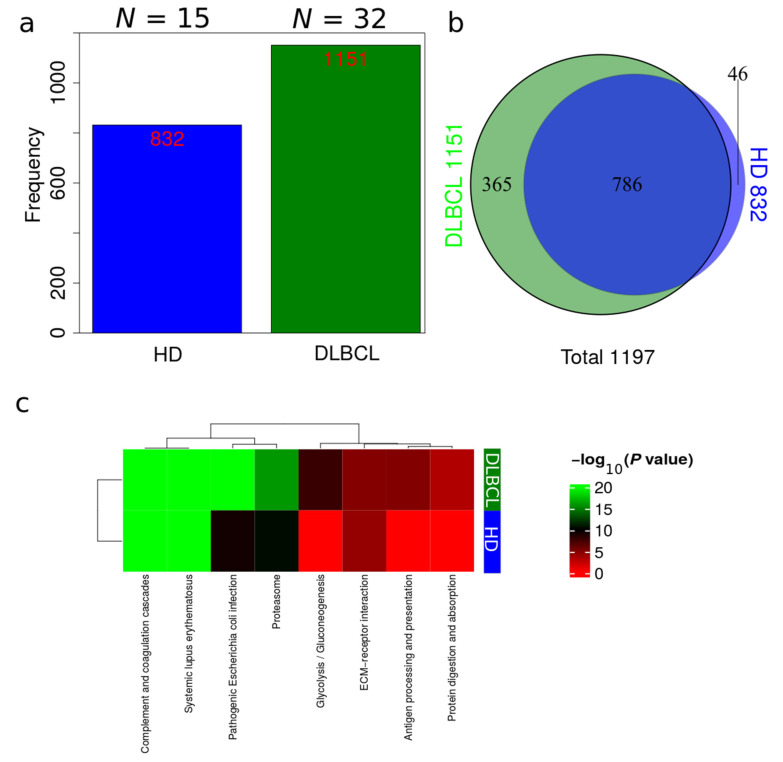
Overview of identifications in DLBCL patients and HDs. (**a**) Total number of identifications from DLBCL patients and HDs. (**b**) Venn diagram comparing identifications in DLBCL patients and HDs. (**c**) Summary of functional KEGG pathway analysis that are either enriched in HDs or DLBCL patients based on all identified proteins from DLBCL and HD groups.

**Figure 4 ijms-23-13327-f004:**
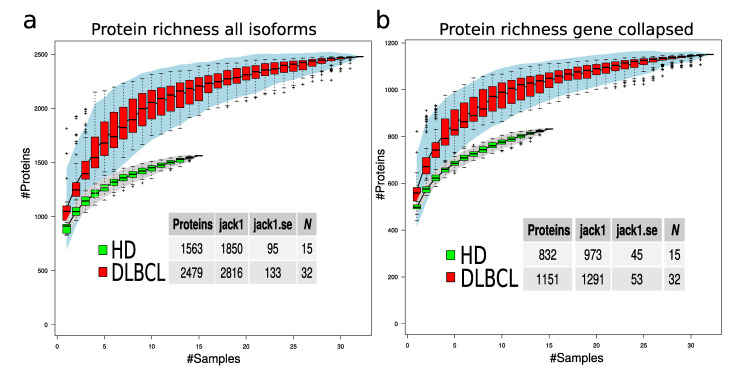
Protein richness in DLBCL and HD groups. (**a**) Protein richness curves for DLBCL (red boxes) and HD group (green boxes) when all protein isoforms are used. (**b**) Protein richness curves for DLBCL (red boxes) and HD group (green boxes) when gene-collapsed proteins were used. The lower tables indicate the estimated number of proteins in the two sample groups using different estimation methods and the corresponding standard error.

**Figure 5 ijms-23-13327-f005:**
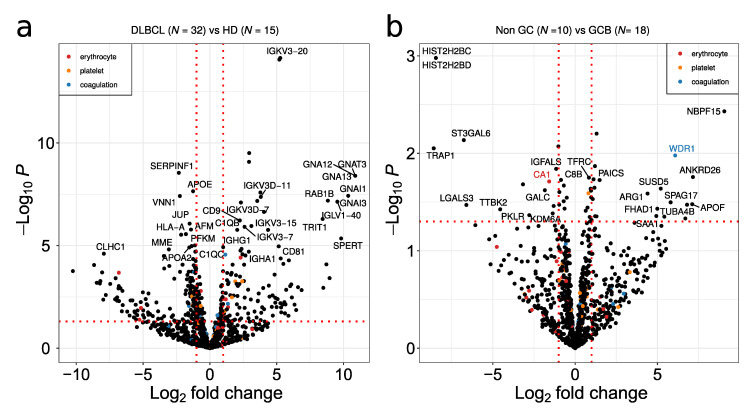
Volcano plot depicting deregulated proteins when comparing (**a**) DLBCL versus HD and (**b**) non-GCB versus GCB. A *p*-value threshold of 0.05 is indicated by a horizontal red dotted line. The log_2_ fold up or down regulation is represented by the red vertical line. Potential contaminants from erythrocyte, platelet and coagulation are indicated by red, orange and blue dots, respectively.

**Figure 6 ijms-23-13327-f006:**
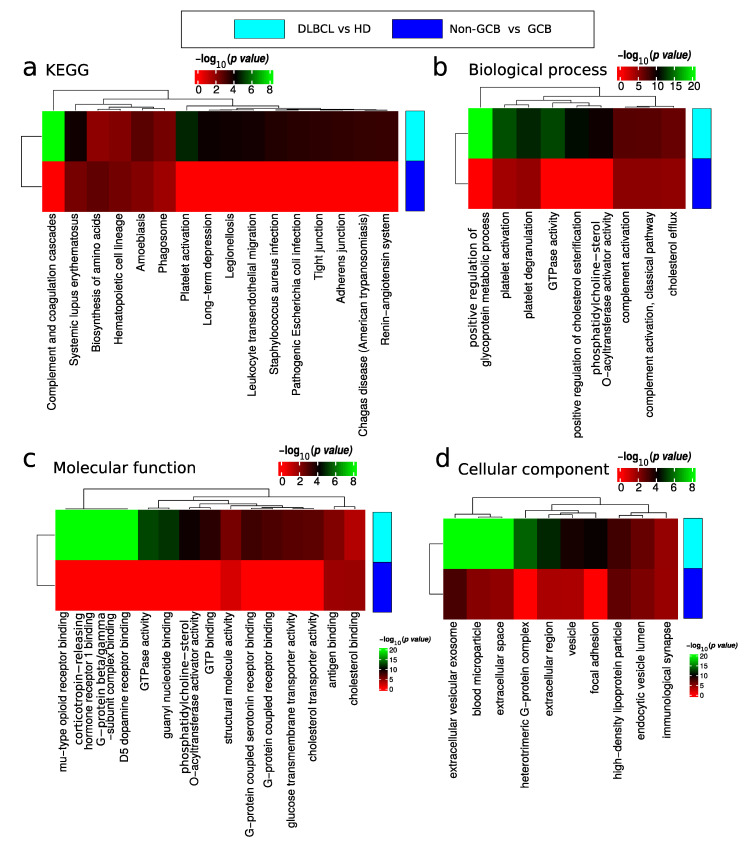
Functional enrichment analysis of differentially expressed proteins between DLBCL and HD and non-GCB and GCB. Analysis of (**a**) KEGG pathway, (**b**) biological process, (**c**) molecular function and (**d**) cellular component gene categories.

**Figure 7 ijms-23-13327-f007:**
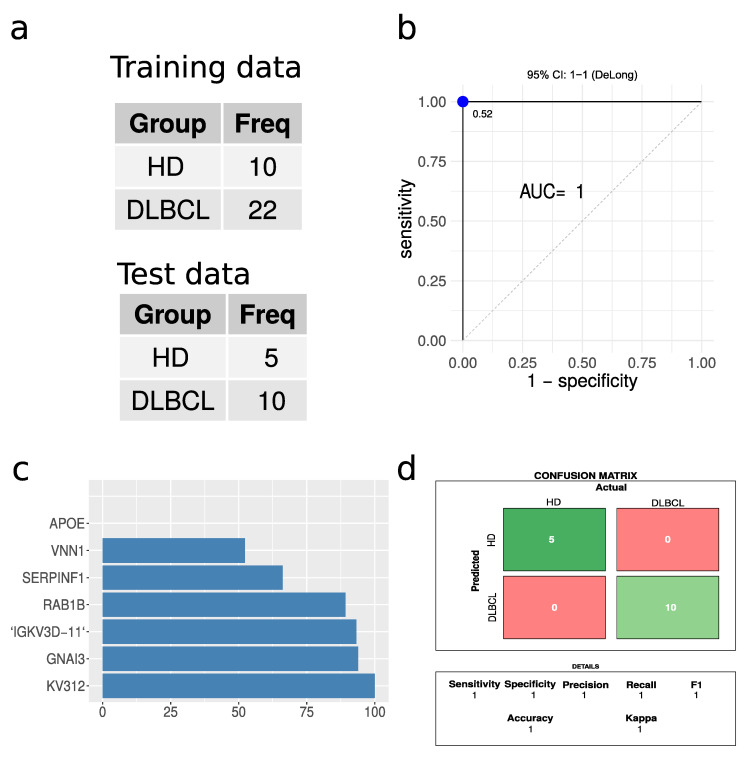
Performance of partial least squares regression analysis. (**a**) A summary of the training and test data sets used. (**b**) Receiver operating characteristics for the PLS regression model. The blue dot indicates the threshold based on Youden’s J statistic. (**c**) The top ten protein features in terms of importance. (**d**) The confusion matrix and the performance parameters associated with it.

**Figure 8 ijms-23-13327-f008:**
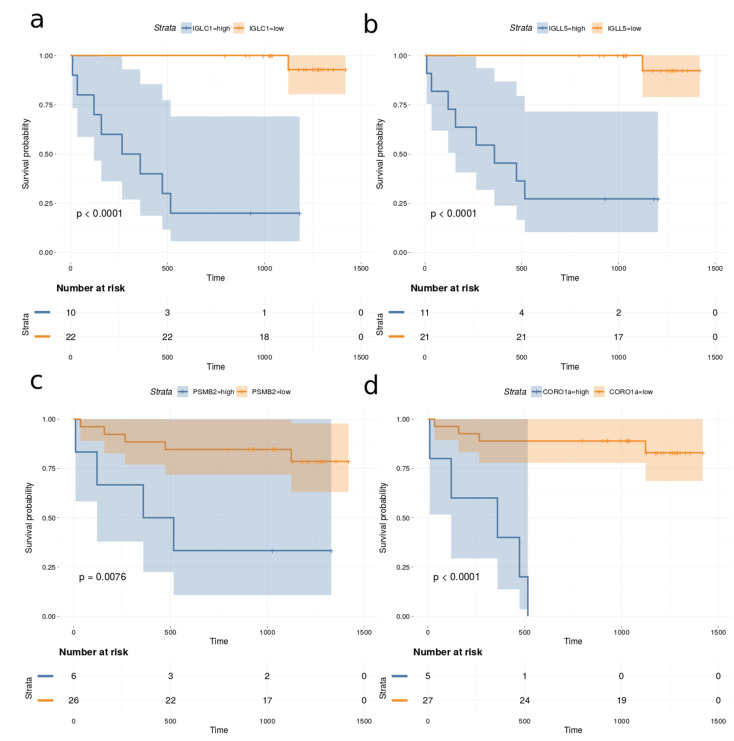
Kaplan–Meier plots based on dividing high and low expression of (**a**) IGLC1, (**b**) IGLL5, (**c**) PSMB2 and (**d**) CORO1a.

**Table 1 ijms-23-13327-t001:** Patients demographics.

Characteristics	*N*	% or Mean (Range)
Sex		
Male/Female	16/16	50%
**Age (years)**	32	67.2 (30–94)
**LDH ^1^**		
≤250	19	180 (100–241)
>250	13	680 (262–2537)
**IPI score ^2^**		
Low risk = 0/1	10	31.30%
low-intermediate risk = 2	6	18.80%
high-intermediate risk = 3	12	37.5%
high risk = 4/5	4	12.50%
**Ann Arbor stage**		
I	7	21.90%
II	5	15.70%
III	2	6.30%
IV	18	56.30%
**DLBCL subtype ^4^**		
GCB type	21	65.60%
Non-GCB type	10	31.30%
ND ^3^	1	3.10%

^1^ LDH: Lactate dehydrogenase; ^2^ IPI: International prognostic indices; ^3^ ND: Non determined. ^4^ Subtyping based on Hans algorithm.

**Table 2 ijms-23-13327-t002:** Cox–Mantel log-rank tests and *p*-value estimates from Cox proportional hazards regression models.

Protein	*p*	LogRank	Protein	*p*	LogRank
IGLC1	3.31 × 10^−4^	1.21 × 10^−5^	POTEE	2.40 × 10^−3^	3.07 × 10^−4^
IGLL5	3.32 × 10^−4^	1.99 × 10^−5^	POTEI	2.60 × 10^−3^	3.28 × 10^−4^
IGLC6	3.58 × 10^−4^	3.95 × 10^−5^	POTEF	2.65 × 10^−3^	3.74 × 10^−4^
PSMB2	5.18 × 10^−4^	1.98 × 10^−5^	IGLC3	2.73 × 10^−3^	7.07 × 10^−4^
CORO1A	8.72 × 10^−4^	2.84 × 10^−6^	PSMA2	2.85 × 10^−3^	9.08 × 10^−4^
PSMA1	1.01 × 10^−3^	4.16 × 10^−6^	IGHV3_16	3.42 × 10^−3^	4.72 × 10^−4^
ARPC3	1.30 × 10^−3^	1.76 × 10^−5^	TUBA3C	3.46 × 10^−3^	1.40 × 10^−4^
HLA_E	1.42 × 10^−3^	2.05 × 10^−4^	ACTBL2	3.63 × 10^−3^	1.31 × 10^−5^
PSMB4	1.50 × 10^−3^	1.38 × 10^−4^	FCHO2	3.70 × 10^−3^	3.57 × 10^−4^
PON1	1.88 × 10^−3^	1.01 × 10^−4^	TUBA4A	4.05 × 10^−3^	1.81 × 10^−4^
ACTC1	1.98 × 10^−3^	4.90 × 10^−5^	SERPING1	4.06 × 10^−3^	1.53 × 10^−3^
ACTA1	1.98 × 10^−3^	4.90 × 10^−5^	TUBA1C	4.10 × 10^−3^	9.24 × 10^−5^
ACTG1	2.03 × 10^−3^	3.94 × 10^−5^	TUBA1A	4.17 × 10^−3^	9.37 × 10^−5^
ACTB	2.12 × 10^−3^	6.31 × 10^−5^	ENO3	4.32 × 10^−3^	1.12 × 10^−3^
ACTA2	2.32 × 10^−3^	7.35 × 10^−5^	TUBA1B	4.98 × 10^−3^	1.51 × 10^−4^
ACTG2	2.32 × 10^−3^	7.35 × 10^−5^			

## Data Availability

The mass spectrometry proteomics data have been deposited to the ProteomeXchange Consortium via the PRIDE [62] partner repository with the dataset identifier PXD036735.

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
