# Peer review of "Extracellular Vesicles in Diffuse Large B Cell Lymphoma: Characterization and Diagnostic Potential"

_ijms, 2022, doi:10.3390/ijms232113327_

Round 1
Reviewer 1 Report
In their paper, the Authors provide a proof of concept for the application of proteins EVs as a set of biomarkers for the detection of DLBCL in patient's plasma. Even though the number of patients was not very high in the present study, undoubtedly the use of their methodology allowed differentiation of DLBCL from HD. As for the specific proteins identified in this study with potential use for diagnostics, the G-proteins and immunoglobulins are the most promising targets.
Overall the work is properly done and presented in a clear and concise fashion. The quality of the figures is appropriate for this journal. The figure's captions properly describe the contents. The material and methods section provides sufficiently good description of used methodology
I could not find the supplementary Table 1 in appendix A
Author Response
We thank the reviewers for the positive comments and points for improvement. We consider that the revised manuscript has further increased in impact, given that the analysis of survival identified several proteins significantly correlated with survival. We have replied to all the reviewers' concerns below.
Reviewer 2 Report
Thank you for this well written manuscript that very nicely describes the characterisation of protein signatures of plasma-isolated exosomes from patients diagnosed with DLBCL, shown here allow distinction from that identified from healthy donors.
The manuscript would greatly benefit from defining the composition of DLBCL cohort with reference to location of the DLBCL at first diagnosis; namely, systemic and/or central nervous involvement (CNS). This is of interest, and would provide useful information, given the differing prognostic expectations and survival outcomes for DLBCL in these two locations associated with differing diagnostic approaches and challenges, and/or treatment efficacy, and/or prevalence of cell-of-origin types. This may only require minor typographical updates in the manuscript to provide clarification.
However, should the cohort include both systemic and CNS primary diagnoses of DLBCL, it is recommended that the manuscript includes comment about how this may relate to observations reported - for example; proteome diversity and variance in identified proteins across the DLBCL cohort as it may potentially relate to CNS and non-CNS DLBCL involvement within each cell-of-origin grouping - and how this may inform future research approaches (without the requirement of further analyses on this potential paired grouping within the DLBCL cohort).
Figure 4 a & b may benefit from inclusion of a coloured green and red square by the inset table labels for 'HD' and 'DLBCL', respectively, to provide visual reference associated graphed data.
In general figures and tables would present better if the same font was used for labels across all, and the font size was consistent for figure sub-part labels (ie; parts a, b, c, etc.).
Kind regards
Author Response
AUs:
We thank the reviewers for the positive comments and points for improvement. We consider that the revised manuscript has further increased in impact, given that the analysis of survival identified several proteins significantly correlated with survival. We have replied to all the reviewers' concerns below.
R:
The manuscript would greatly benefit from defining the composition of DLBCL cohort with reference to location of the DLBCL at first diagnosis; namely, systemic and/or central nervous involvement (CNS). This is of interest, and would provide useful information, given the differing prognostic expectations and survival outcomes for DLBCL in these two locations associated with differing diagnostic approaches and challenges, and/or treatment efficacy, and/or prevalence of cell-of-origin types. This may only require minor typographical updates in the manuscript to provide clarification.
However, should the cohort include both systemic and CNS primary diagnoses of DLBCL, it is recommended that the manuscript includes comment about how this may relate to observations reported - for example; proteome diversity and variance in identified proteins across the DLBCL cohort as it may potentially relate to CNS and non-CNS DLBCL involvement within each cell-of-origin grouping - and how this may inform future research approaches (without the requirement of further analyses on this potential paired grouping within the DLBCL cohort).
AUs:
This is a very interesting point and could be a very interesting follow-up study. We add the following statement to the manuscript: "At the time of diagnosis, only one patient presented with central nervous system involvement".
R:
Figure 4 a & b may benefit from inclusion of a coloured green and red square by the inset table labels for 'HD' and 'DLBCL', respectively, to provide visual reference associated graphed data.
AUs:
We included a colored green and red square to further clarify the sample groups.
R:
In general, figures and tables would present better if the same font was used for labels across all, and the font size was consistent for figure sub-part labels (ie; parts a, b, c, etc.).
AUs:
Font size in the figure sub-part labels were standardized to “san serif size 36”.